# SAPA: Similarity-Aware Point Affiliation for Feature Upsampling

**Hao Lu**   **Wenze Liu**   **Zixuan Ye**   **Hongtao Fu**   **Yuliang Liu**   **Zhiguo Cao**[*]
School of Artificial Intelligence and Automation
Huazhong University of Science and Technology
Wuhan 430074, China
`{hlu,zgcao}@hust.edu.cn`

## Abstract

We introduce point affiliation into feature upsampling, a notion that describes the affiliation of each upsampled point to a semantic cluster formed by local decoder feature points with semantic similarity. By rethinking point affiliation, we present a generic formulation for generating upsampling kernels. The kernels encourage not only semantic smoothness but also boundary sharpness in the upsampled feature maps. Such properties are particularly useful for some dense prediction tasks such as semantic segmentation. The key idea of our formulation is to generate similarity-aware kernels by comparing the similarity between each encoder feature point and the spatially associated local region of decoder features. In this way, the encoder feature point can function as a cue to inform the semantic cluster of upsampled feature points. To embody the formulation, we further instantiate a lightweight upsampling operator, termed Similarity-Aware Point Affiliation (SAPA), and investigate its variants. SAPA invites consistent performance improvements on a number of dense prediction tasks, including semantic segmentation, object detection, depth estimation, and image matting. Code is available at: `https://github.com/poppinace/sapa`

## 1 Introduction

We introduce the notion of point affiliation into feature upsampling. Point affiliation defines a relation between each upsampled point and a *semantic cluster*[2] to which the point should belong. It highlights the spatial arrangement of upsampled points at the semantic level. Considering an example shown in Fig. 1 w.r.t. $\times 2$ upsampling in semantic segmentation, the orange point of low resolution will correspond to four upsampled points of high resolution, in which the red and yellow ones should be assigned the 'picture' cluster and the 'wall' cluster, respectively. Designating point affiliation is difficult and sometimes can be erroneous, however.

In $\times 2$ upsampling, nearest neighbor (NN) interpolation directly copies four identical points from the low-res one, which assigns the same semantic cluster to the four points. On regions in need of details, the four points probably do not share the same cluster but are forced to share. Bilinear interpolation assigns point affiliation with distance priors. Yet, when tackling points of different semantic clusters, it not only cannot inform clear point affiliation, but also blurs the boundary between different semantic clusters. Recent dynamic upsampling operators have similar issues. CARAFE [1, 2] judges the affiliation of an upsampled point with content-aware kernels. Certain semantic clusters will receive larger weights than the rest and therefore dominate the affiliation of upsampled points. However, the affiliation near boundaries or on regions full of details can still be ambiguous. As shown in Fig. 2, the

---

[*]Corresponding author.

[2]A semantic cluster is formed by local decoder feature points with the similar semantic meaning.

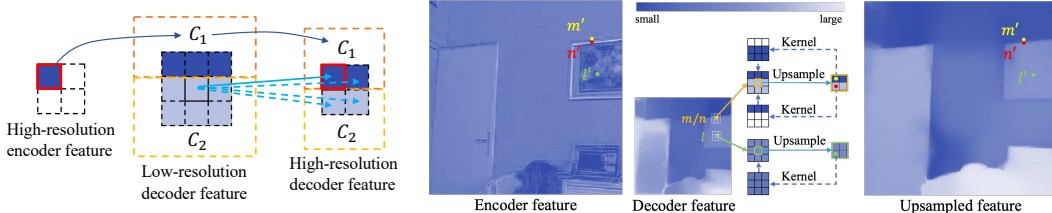

Figure 1: **Left: Similarity between an encoder point and different semantic clusters in the decoder. Right: Point affiliation mechanism of ideal upsampling kernels.** Left: If the red-box encoder feature point is classified into the semantic cluster $C_1$, then it is more similar to $C_1$ than $C_2$. Right: The upsampling kernels can be a 'soft' selector in a local window to assign point affiliation. For an upsampled point, the kernel selects a/some representative points from its most relative semantic cluster. *E.g.*, according to the encoder feature, the red upsampled feature point should belong to the 'picture' cluster. Then we expect the kernel can assign large weights on picture-related points and small weights on wall-related points. In this way, after the weighted sum, the upsampled point will be revalued and assigned the 'picture' cluster.

boundaries are unclear in the feature maps upsampled by CARAFE. The reason is that the kernels are conditioned on decoder features alone; the decoder features carry little useful information about high-res structure.

Inferring structure requires high-res encoder features. For instance, if the orange point in Fig. 1 lies on the low-res boundary, it is difficult to judge to which cluster the four upsampled points should belong. However, the encoder feature in Fig. 1 actually tells that, the yellow point is on the wall, and the red one is on the picture, which suggests one may extract useful information from the encoder feature to assist the designation of point affiliation. Indeed IndexNet [3, 4] and A2U [5] have attempted to encode such information to improve detail delineation in encoder-dependent upsampling kernels; however, the encoder feature can easily introduce noise into kernels, engendering discontinuous feature maps shown in Fig. 2. Hence, the key problem seems to be how to extract only required information into the upsampling kernels from the encoder feature while filtering out the rest.

To use encoder features effectively, an important assumption of this paper is that, *an encoder feature point is most similar to the semantic cluster into which the point will be classified.* Per the left of Fig. 1, suppose that the encoder point in the red box is assigned into the cluster $C_1$ by its semantic meaning, then it is similar to $C_1$, while not similar to $C_2$. As a result, by comparing the similarity between the encoder feature point and different semantic clusters in the decoder feature, the affiliation of the upsampled point can be informed according to the similarity scores. In particular, we propose to generate upsampling kernels with local mutual similarity between encoder and decoder features. For every encoder feature point, we compute the similarity score between this point and each decoder feature point in the spatially associated local window. For the green point in Fig. 1, since every point in the window shares the same semantic cluster, the encoder feature point is as similar as every point in the window. In this case we expect an 'average kernel' which is the key characteristic to filter noise, and the upsampled four points would have the same semantic cluster as before. For the yellow point in the encoder, since it belongs to the 'wall' cluster, it is more similar to the points on the wall than those on the picture. In this case we expect a kernel with larger weights on points related to the 'wall' cluster. This can help to assign the affiliation of the yellow point to be in the 'wall' cluster.

By modeling the local mutual similarity, we derive a generic form of upsampling kernels and show that this form implements our expected upsampling behaviors: encouraging both semantic smoothness and boundary sharpness. Following our formulation, we further instantiate a lightweight upsampling operator, termed Similarity-Aware Point Affiliation (SAPA), and investigate its variants. We evaluate SAPA across a number of mainstream dense prediction tasks, for example: i) *semantic segmentation*: we test SAPA on several transformer-based segmentation baselines on the ADE20K dataset [6], such as SegFormer [7], MaskFormer [8], and Mask2Former [9], improving the baselines by $1\% \sim 2.7\%$ mIoU; ii) *object detection*: SAPA improves the performance of Faster R-CNN by $0.4\%$ AP on MS COCO [10]; iii) *monocular depth estimation*: SAPA reduces the rmse metric of BTS [11] from $0.419$ to $0.408$ on NYU Depth V2 [12]; and iv) *image matting*: SAPA outperforms a strong A2U matting baseline [5] on the Adobe Composition-1k testing set [13] with a further $3.8\%$ relative error reduction in the SAD metric. SAPA also outperforms or at least is on par with other state-of-the-art

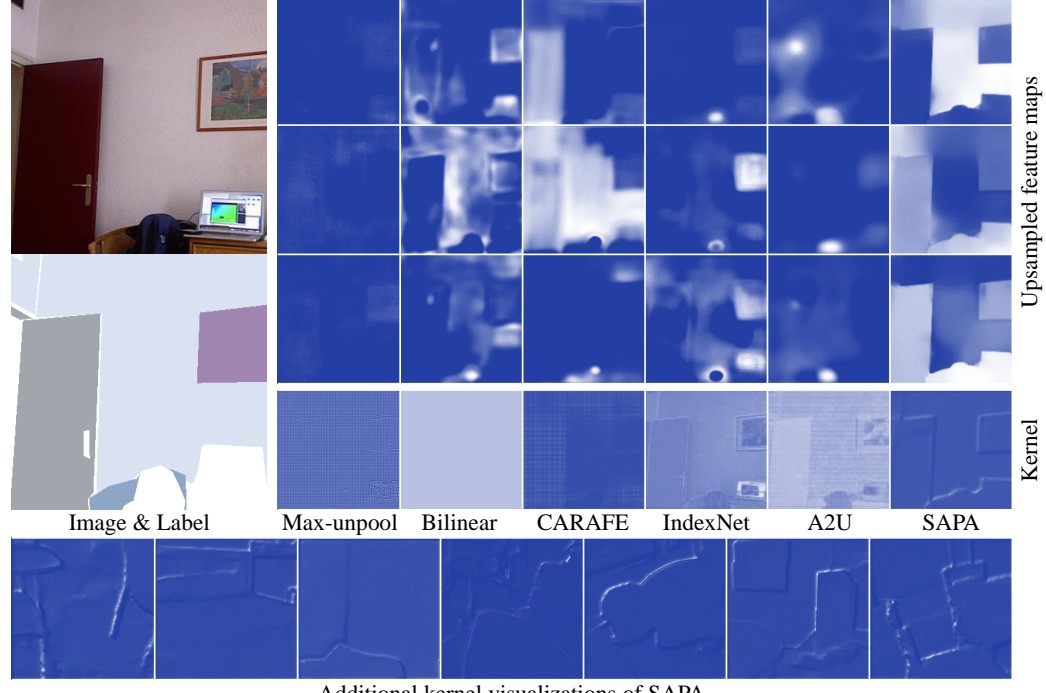

Figure 2: **Top: Upsampled feature maps and upsampling kernel maps of different upsampling operators. Down: Additional upsampling kernel maps of SAPA.** The visualizations are produced with SegNet [14] as the baseline on the SUN RGBD [15] dataset. For each upsampling operator, we choose the first three channels from the feature maps of the last upsampling stage for visualization. Only SAPA shows both smooth regions and sharp boundaries. The kernel map of CARAFE is coarse and lacking in details, while IndexNet and A2U generate kernels with much undesired details from the encoder. See the supplementary material for additional visualizations.

dynamic upsampling operators. Particularly, even without additional parameters, SAPA outperforms the previous best upsampling operator CARAFE on semantic segmentation.

## 2   Related Work

We review work related to feature upsampling. Feature upsampling is a fundamental procedure in encoder-decoder architectures used to recover the spatial resolution of low-res decoder feature maps and has been extensively used in dense prediction tasks such as semantic segmentation [14, 7, 8, 9] and depth estimation [16, 17, 11].

Standard upsampling operators are hand-crafted. NN and bilinear interpolation measure the semantic affiliation in terms of relative distances in upsampling, which follows fixed rules to designate point affiliation, even if the true affiliation may be different. Max unpooling [14] stores the indices of max-pooled feature points in encoder features and uses the sparse indices to guide upsampling. While it executes location-specific point affiliation which benefits detail recovery, most upsampled points are assigned with null affiliation due to zero filling. Pixel Shuffle [18] is widely-used in image/video super-resolution. Its upsampling only includes memory operation – reshaping depth channels to space ones. The notion of point affiliation does not apply to this operator, however.

Another stream of upsampling operators implement learning-based upsampling. Among them, transposed convolution or deconvolution [19] is known as an inverse convolutional operator. Based on a novel interpretation of deconvolution, PixelTCL [20] is proposed to alleviate the checkerboard artifacts [21] of standard deconvolution. In addition, bilinear additive upsampling [22] attempts to combine learnable convolutional kernels with hand-crafted upsampling operators to achieve composited upsampling. Recently, DUpsample [23] seeks to reconstruct the upsampled feature map with pre-learned projection matrices, expecting to achieve a data-dependent upsampling behavior.

While these operators are learnable, the upsampling kernels are fixed once learned, still resulting in ambiguous designation of point affiliation.

In learning-based upsampling, some recent work introduces the idea of generating content-aware dynamic kernels. Instead of learning the parameters of the kernels, they learn how to predict the kernels. In particular, CARAFE [1, 2] predicts dynamic kernels conditioned on the decoder features. IndexNet [3, 4] and A2U [5], by contrast, generate encoder-dependent kernels. While they significantly outperform previous upsampling operators in various tasks, they still can cause uncertain point affiliation, resulting in either unclear predictions near boundaries or fragile predictions in regions.

Our work is closely related to dynamic kernel-based upsampling. We also seek to predict dynamic kernels; however, we aim to address the uncertain point affiliation in prior arts to achieve simultaneous region smoothness and boundary sharpness.

## 3 Dynamic Upsampling Revisited

We first revisit two key components shared by existing dynamic upsampling operators: kernel generation and feature assembly.

**Kernel Generation** Given the decoder feature $\mathcal{X} \in \mathbb{R}^{H \times W \times C}$, if we upsample it to the target feature $\mathcal{X}' \in \mathbb{R}^{2H \times 2W \times C}$, then for any point at the location $l' = (i', j')$ in $\mathcal{X}'$, we generate a kernel $\mathcal{W}_{l'}$ based on the neighborhood feature $\mathcal{N}_{l'}$ that spatially corresponds to $l'$. In this way, kernel generation can be defined by

$$\mathcal{W}_{l'} = \text{norm}(\psi(\mathcal{N}_{l'})), \tag{1}$$

where $\psi$ refers to a kernel generation module, which is often implemented by a sub-network used to predict the kernel conditioned on $\mathcal{N}_{l'}$, and norm is a normalization function. The feature $\mathcal{N}_{l'}$ can originate from two sources, to be specific, from the encoder feature $\mathcal{Y} \in \mathbb{R}^{2H \times 2W \times C}$ or from the decoder feature $\mathcal{X}$. If the encoder feature $\mathcal{Y}$ is chosen as the source, then a local region of $\mathcal{Y}$ centered at $l'$ is extracted to be $\mathcal{N}_{l'}$. If the decoder feature $\mathcal{X}$ is chosen, one first needs to compute the projective location of $l'$, *i.e.*, $l = (i, j) = \lfloor \frac{l'}{2} \rfloor = (\lfloor \frac{i'}{2} \rfloor, \lfloor \frac{j'}{2} \rfloor)$ according to the spatial location correspondence, then a local region of $\mathcal{X}$ centered at $l$ is regarded as $\mathcal{N}_{l'}$. The softmax function is often used as the normalization function such that relevant points can be softly selected to compute the value of the target point using the weight $\mathcal{W}_{l'}$.

According to Fig. 2, the source of $\mathcal{N}_{l'}$ can affect the predicted kernel. The kernels predicted by CARAFE, IndexNet, and A2U show significantly distinct characteristics. With the decoder feature alone, the kernel map is coarse and lacking in details. Benefiting from the encoder feature, the kernel maps generated by IndexNet and A2U have rich details; however, they manifest high similarity to the encoder feature, which means noise is introduced into the kernel.

**Feature Assembly** For each target feature point at $l'$, we assemble the corresponding sub-region of decoder feature with the predicted $K \times K$ kernel $\mathcal{W}_{l'}$, whose weight is denoted by $\mathcal{W}_{l',p}, p \in I, I = \{(u, v) : u, v = -r, ..., r, r = \lfloor \frac{K}{2} \rfloor\}$, to obtain the value of the target point $\mathcal{X}'_{l'}$ by a weighted sum

$$\mathcal{X}'_{l'} = \sum_{p \in I} \mathcal{W}_{l',p} \mathcal{X}_{l+p}. \tag{2}$$

By executing feature assembly on every target feature point, we can obtain the target upsampled feature map. As shown in Fig. 2, the upsampled feature has a close relation to the kernel. A well-predicted kernel can encourage both semantic smoothness and boundary sharpness; a kernel without encoding details or with too many details encoded can introduce noise. We consider an ideal kernel should only response at the position in need of details, while do not response (appearing as an average value over an area) at good semantic regions. More importantly, an ideal kernel should assign weights reasonably so that each point can be designated to a correct semantic cluster.

## 4 Rethinking Point Affiliation with Local Mutual Similarity

To obtain an expected upsampling kernel mentioned above, we first derive a generic formulation for generating the upsampling kernel by exploiting local mutual similarity, then explain why the formula-

tion encourages semantic smoothness and boundary sharpness, and finally present an upsampling operator, SAPA, that embodies our formulation.

## 4.1 Local Mutual Similarity

We rethink point affiliation from the view of local mutual similarity between encoder and decoder features. With a detailed analysis, we explain why such similarity can assist point affiliation.

We first define a generic similarity function $\text{sim}(\boldsymbol{x}, \boldsymbol{y}) : \mathbb{R}^d \times \mathbb{R}^d \to \mathbb{R}$. It scores the similarity between a vector $\boldsymbol{x}$ and a vector $\boldsymbol{y}$ of the same dimension $d$. We also define a normalization function involving $n$ real numbers $x_1, x_2, ..., x_n$ by $\text{norm}(x_i; x_1, x_2, ..., x_n) = \frac{h(x_i)}{\sum_{j=1}^{n} h(x_j)}$, where $h(x) : \mathbb{R} \to \mathbb{R}$ is an arbitrary function, ignoring zero division. Given $\text{sim}(\boldsymbol{x}, \boldsymbol{y})$ and $h(x)$, we can define a generic formulation for generating the upsampling kernel

$$w = \frac{h\left(\text{sim}(\boldsymbol{x}, \boldsymbol{y})\right)}{\sum\limits_{\boldsymbol{x}' \in \mathcal{N}_{l'}} h\left(\text{sim}(\boldsymbol{x}', \boldsymbol{y})\right)} \,, \tag{3}$$

where $w$ is the kernel value specific to $\boldsymbol{x}$ and $\boldsymbol{y}$. To analyze the upsampling behavior of the kernel, we further define the following notations.

Let $\boldsymbol{y}_{l'} \in \mathbb{R}^C$ denote the encoder feature vector at position $l'$ and $\boldsymbol{x}_l \in \mathbb{R}^C$ be the decoder feature vector at position $l$, where $C$ is the number of channels. Our operation will be done within a local window of size $K \times K$, between each encoder vector $\boldsymbol{y}_{l'}$ and all its spatially associated decoder feature vectors, $\boldsymbol{x}_{l+p}$'s.

To simplify our analysis, we also assume local smoothness. That is, points with the same semantic cluster will have a similar value, which means a local region will share the same value on every channel of the feature map. As shown in Fig. 1, we choose $\boldsymbol{a}$ and $\boldsymbol{b}$ as the feature vectors in 'wall' and 'picture' cluster, respectively, then $\boldsymbol{a}$ and $\boldsymbol{b}$ are constant vectors. For ease of analysis, we define two types of windows distinguished by their contents. When all the points inside a window belong to the same semantic cluster, it is called a smooth window; while different semantic clusters appear in a window, it is defined as a detail window.

Next, we explain why the kernel can filter out noise, why it encourages semantic smoothness in a smooth window, and why it can help to recover details when dealing with boundaries/textures in a detail window.

**Upsampling in a Smooth Window**    Without loss of generality, we consider an encoder point at the position $l'$ in Fig. 1. Its corresponding window is a smooth window of the semantic cluster 'picture', thus $\boldsymbol{x}_{l+p} = \boldsymbol{b}, \forall p \in I$. Then the upsampling kernel weight w.r.t. the upsampled point $l'$ at the position $p$ takes the form

$$\mathcal{W}_{l',p} = \text{norm}(\text{sim}(\boldsymbol{x}_{l+p}, \boldsymbol{y}_{l'})) = \frac{h(\text{sim}(\boldsymbol{x}_{l+p}, \boldsymbol{y}_{l'}))}{\sum\limits_{q \in I} h(\text{sim}(\boldsymbol{x}_{l+q}, \boldsymbol{y}_{l'}))} = \frac{h(\text{sim}(\boldsymbol{b}, \boldsymbol{y}_{l'}))}{K^2 h(\text{sim}(\boldsymbol{b}, \boldsymbol{y}_{l'}))} = \frac{1}{K^2} \,, \tag{4}$$

which has nothing to do with $l$, and $p$. Eq. (4) reveals a key characteristic of local mutual similarity in a smooth window: the kernel weight is a constant regardless of $\boldsymbol{y}$. Therefore, the kernel fundamentally can filter out noise from encoder features with an 'average' kernel.

Note that, in the derivation above the necessary conditions include: i) $\boldsymbol{x}$ is from a local window in the decoder feature map; ii) a normalization function in the form of $\frac{h(x_i)}{\sum_j h(x_j)}$.

**Upsampling in a Detail Window**    Again we consider two encoder points at the position $m'$ and $n'$ in Fig. 1. Ideally $\boldsymbol{y}_{m'}$ and $\boldsymbol{y}_{n'}$ should be classified into the semantic cluster of 'wall' and 'picture', respectively. Taking $\boldsymbol{y}_{m'}$ as an example, following our assumption, it is more similar to points of the 'wall' cluster rather than the 'picture' cluster. From Eq. (4), we can tell that $\text{sim}(\boldsymbol{x}_{m+s}, \boldsymbol{y}_{m'})$ is larger than $\text{sim}(\boldsymbol{x}_{m+t}, \boldsymbol{y}_{m'})$, where $s$ and $t$ are the offsets in the window such that $m + s$ and $m + t$ are within the 'wall' and the 'picture' cluster, respectively. Therefore, after computing similarity scores and normalization, one can acquire a kernel with significantly larger weights on points with the semantic cluster of 'wall' than that of 'picture', *i.e.*, $\mathcal{W}_{m',s} >> \mathcal{W}_{m',t}$. After applying the kernel to the corresponding window, the upsampled point at $m'$ will be revalued and assigned to the semantic cluster of 'wall'. Similarly, the upsampled point at $n'$ will be assigned into 'picture'.

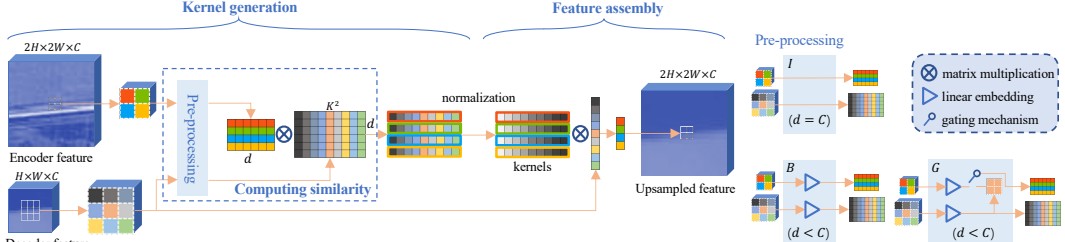

Figure 3: **Feature upsampling of SAPA.** We compute mutual similarity scores with a similarity function between each encoder feature point and the spatially associated local decoder feature points (features are normalized with `LayerNorm` before similarity computation). The scores are then transformed into upsampling kernel weights after kernel normalization. The kernel is then used to assemble the upsampled feature points. We illustrate three variants: SAPA-I with inner-product similarity, SAPA-B with (low-rank) bilinear similarity, and SAPA-G with gated (low-rank) bilinear similarity. They differ in the pre-processing step. See detailed definition in Section 4.2.

Note that, in Eq. (4) we have no constraint on $\boldsymbol{y}$. But here in a detail window, $\boldsymbol{y}$ as an encoder feature vector can play a vital role for designating correct point affiliation. Next in our concrete implementation, we discuss how to appropriately use $\boldsymbol{y}$ in the similarity function.

## 4.2 SAPA: Similarity-Aware Point Affiliation

Here we embody Eq. (3) by investigating different similarity and normalization functions.

**Normalization Function**   Though we do not constrain $h(x)$ in theory, in reality it must be carefully chosen. For example, to avoid zero division, the scope for the choice of $h(x)$ is narrowed. Following existing practices [1, 3, 5], we choose $h(x) = e^x$ by default, which is equivalent to `softmax` normalization. We also test some other $h(x)$'s, such as $h(x) = \texttt{ReLU}(x)$, $h(x) = \texttt{sigmoid}(x)$, and $h(x) = \texttt{softplus}(x)$. Their performance comparisons will be given in ablation studies.

**Similarity Function**   We study three types of similarity functions:

- Inner-product similarity: $\text{sim}(\boldsymbol{x}, \boldsymbol{y}) = \boldsymbol{x}^T \boldsymbol{y}$,
- (Low-rank) bilinear similarity [24]: $\text{sim}(\boldsymbol{x}, \boldsymbol{y}) = \boldsymbol{x}^T P_{\boldsymbol{x}}^T P_{\boldsymbol{y}} \boldsymbol{y}$,
- Gated (low-rank) bilinear similarity [25]: $\text{sim}(\boldsymbol{x}, \boldsymbol{y}) = g \boldsymbol{x}^T P_{\boldsymbol{x}}^T P_{\boldsymbol{xy}} \boldsymbol{y} + (1-g) \boldsymbol{x}^T P_{\boldsymbol{x}}^T P_{\boldsymbol{xx}} \boldsymbol{x}$,

where $P_{\boldsymbol{x}} \in \mathbb{R}^{d \times C}$, $P_{\boldsymbol{y}} \in \mathbb{R}^{d \times C}$, $P_{\boldsymbol{xy}} \in \mathbb{R}^{d \times C}$, and $P_{\boldsymbol{xx}} \in \mathbb{R}^{d \times C}$ are the linear projection matrices, $d$ is the embedding dimension, and $g \in (0, 1)$ is a gate unit learned by linear projection. We model the projection matrices to be low-rank, *i.e.*, $d < C$, to reduce the number of parameters and computational complexity. The gate-modulated bilinear similarity is designed to further filter out the encoder noise. The gate is generated by learning a linear projection matrix that projects the decoder feature $\mathcal{X}$ to a single-channel mask, followed by `sigmoid` normalization. Finally, we have $\mathcal{Y} = G\mathcal{Y} + (1-G)\mathcal{X}$ (nearest neighbor interpolation is used for matching the resolution), where $G$ is the matrix form of the gate unit. The use of the gate implies we reduce noise in some spatial regions by replacing encoder features $\mathcal{Y}$ with decoder features $\mathcal{X}$. Its vector form explains that in the area of noise it tends to switch to the self-similarity mode. We will prove the effectiveness of the gating mechanism by comparing the gate-modulated bilinear similarity with a baseline without the gating mechanism.

In practice, encoder and decoder features have different data distributions, which is unsuitable to compare similarity directly. Therefore `LayerNorm` is applied to encoder and decoder features before similarity computation. Indeed we observe that the loss does not decrease without normalization.

As shown in Fig. 3, our implementation is similar to the previous dynamic upsampling operators, which first generates the upsampling kernels and then assembles the decoder feature conditioned on the kernels. We highlight the kernel generation module. By setting a kernel size of $K$, for each encoder feature point, we compute the similarity scores between this point and each of $K \times K$ neighbors in the decoder. Then, the `softmax` normalization is applied to generate the upsampling kernels. SAPA is lightweight and can even work without additional parameters (with inner-product similarity).

Table 1: Computational complexity and parameters of SAPA and other upsamplers. $C$: number of channels, $d$: embedded channel number. As the default settings, $d = 64$, $K = 5$ in CARAFE and $K = 3$ in A2U. We set $d = 32$ and $K = 5$ in SAPA. **I**: inner-product similarity; **B**: bilinear similarity; **G**: gated bilinear similarity.

| Module | Operation | FLOPs ($\times HW$) | Params |
|---|---|---|---|
| CARAFE | Kernel generation | $Cd + 36K^2d$ | $Cd + 36K^2d$ |
| | Feature assembly | $4K^2C$ | 0 |
| | **Total** | $Cd + 36K^2d + 4K^2C$ | $Cd + 36K^2d$ |
| IndexNet | Kernel generation | $32C^2 + 8C$ | $32C^2 + 8C$ |
| HIN | Feature assembly | $4C$ | 0 |
| | **Total** | $32C^2 + 12C$ | $32C^2 + 8C$ |
| IndexNet | Kernel generation | $68C^2$ | $68C^2$ |
| M2O | Feature assembly | $4C$ | 0 |
| | **Total** | $68C^2 + 4C$ | $68C^2$ |
| A2U | Kernel generation | $73C + 4K^2$ | $4K^2C + 2C$ |
| | Feature assembly | $4K^2C$ | 0 |
| | **Total** | $73C + 4K^2 + 4K^2C$ | $4K^2C + 2C$ |
| SAPA | (1) Feature embedding | $5Cd$ | $2Cd$ |
| | (2) Gated addition | $C + 8d$ | $C$ |
| | (3) Inner product | $4K^2d$ | 0 |
| | (4) Feature assembly | $4K^2C$ | 0 |
| | **I Total** (3)(4) ($d = C$) | $8K^2C$ | 0 |
| | **B Total** (1)(3)(4) | $5Cd + 4K^2d + 4K^2C$ | $2Cd$ |
| | **G Total** (1)(2)(3)(4) | $5Cd + 4K^2d + 4K^2C + C + 5d$ | $2Cd + C$ |

To intuitively understand its lightweight property, we compare the computational complexity and number of parameters of different dynamic upsampling operators in Table 1. For example, when upsampling a 256-channel feature map, the FLOPs are $H * W * 99K$, $H * W * 2M$, $H * W * 28K$, and $H * W * 70K$ for CARAFE, IndexNet-HIN, A2U, and SAPA-B, respectively.

We visualize the feature maps of upsampling kernels and upsampled features in Fig. 2. Our upsampling kernels show more favorable responses than other upsampling operators, with weights highlighted on boundaries and noise suppressed in regions, which visually supports our proposition and is a concrete embodiment of Eq. (4).

## 5 Experiments

We first focus our experiments on semantic segmentation to justify the effectiveness of SAPA. We then showcase its universality across three additional dense prediction tasks, including object detection, depth estimation, and image matting. All our experiments are run on a server with 8 NVIDIA GeForce RTX 3090 GPUs.

### 5.1 Data Sets, Metrics, Baselines, and Protocols

For semantic segmentation, we conduct experiments on the ADE20K dataset [6] and report the mIoU metric. Three strong transformer-based models are adopted as the baselines, including SegFormer-B1 [7], MaskFormer-Swin-Base [8] and Mask2Former-Swin-Base [9], where the Swin-Base backbone is pretrained on ImageNet-22K. All training settings and implementation details are kept the same as the original papers. We only modify the upsampling stages with specific upsampling operators.

For object detection, we use the MS COCO [10] dataset, which involves 80 object categories. We use AP as the evaluation metric. Faster R-CNN [26] with ResNet-50 [27] is adopted as the baseline. We use `mmdetection` [28] and follow the $1\times$ (12 epochs) training configurations.

For depth estimation, we use NYU Depth V2 dataset [12] and its default train/test split. We choose BTS [11] with ResNet-50 as the baseline and follow its training configurations. The standard depth metrics used by previous work is employed to evaluate the performance, including root mean squared error (RMS) and its log version (RMS (log)), absolute relative error (Abs Rel), squared relative error (Sq Rel), average $\log_{10}$ error (log10), and the accuracy with threshold $thr$ ($\delta < thr$). Readers can

Table 2: Semantic segmentation results on ADE20K. **I**: inner-product similarity; **B**: bilinear similarity; **G**: gated bilinear similarity. Best performance is in **boldface**, and second best is underlined.

| | SegFormer B1 [7] | | | MaskFormer SwinB [8] | | | Mask2Former SwinB [9] | | |
|---|---|---|---|---|---|---|---|---|---|
| | mIoU↑ | FLOPs | Params | mIoU↑ | FLOPs | Params | mIoU↑ | FLOPs | Params |
| Nearest | – | – | – | 52.70 | 195 | 102 | – | – | – |
| Bilinear | 41.68 | 15.91 | 13.74 | – | – | – | 53.90 | 223 | 107 |
| CARAFE [1] | 42.82 | +1.45 | +0.44 | 53.53 | +0.84 | +0.22 | 53.94 | +0.63 | +0.07 |
| IndexNet [3] | 41.50 | +30.65 | +12.60 | 52.92 | +17.64 | +6.30 | 54.71 | +13.44 | +2.10 |
| A2U [5] | 41.45 | +0.41 | +0.06 | 52.73 | +0.23 | +0.03 | 54.40 | +0.18 | +0.01 |
| SAPA-I | 43.05 | +0.75 | +0.00 | 53.25 | +0.43 | +0.00 | 55.05 | +0.33 | +0.00 |
| SAPA-B | 43.20 | +1.02 | +0.10 | 53.15 | +0.59 | +0.05 | 54.98 | +0.45 | +0.02 |
| SAPA-G | **44.39** | +1.02 | +0.10 | **53.78** | +0.59 | +0.05 | **55.22** | +0.45 | +0.02 |

Table 3: Object detection results with Faster R-CNN on MS COCO. Best performance is in **boldface**, and second best is underlined.

| Faster R-CNN [26] (R50) | Params | $AP$ ↑ | $AP_{50}$ | $AP_{75}$ | $AP_S$ | $AP_M$ | $AP_L$ |
|---|---|---|---|---|---|---|---|
| Nearest | 41.53 | 37.4 | 58.1 | 40.4 | 21.2 | 41.0 | 48.1 |
| CARAFE [1] | +0.22 | **38.6** | **59.9** | **42.2** | **23.3** | **42.2** | **49.7** |
| IndexNet [3] | +6.30 | 37.6 | 58.4 | 40.9 | 21.5 | 41.3 | 49.2 |
| A2U [5] | +0.03 | 37.3 | 58.7 | 40.0 | 21.7 | 41.1 | 48.5 |
| SAPA-I | +0 | 37.7 | 59.2 | 40.6 | 22.2 | 41.2 | 48.4 |
| SAPA-B | +0.05 | 37.8 | 59.2 | 40.6 | 22.4 | 41.4 | 49.1 |
| SAPA-G | +0.05 | 37.8 | 59.1 | 40.8 | 21.5 | 41.4 | 48.6 |

Table 4: Depth estimation results on NYU Depth V2 with BTS. Best performance is in **boldface**.

| BTS [11] | | accuracy ↑ | | | error ↓ | | | | |
|---|---|---|---|---|---|---|---|---|---|
| R50 | Params | $\delta < 1.25$ | $\delta < 1.25^2$ | $\delta < 1.25^3$ | Abs Rel | Sq Rel | RMS | RMS$log$ | log10 |
| Nearest | 49.53 | 0.865 | 0.975 | 0.993 | 0.119 | 0.075 | 0.419 | 0.152 | 0.051 |
| CARAFE [1] | +0.41 | 0.864 | 0.974 | 0.994 | 0.117 | 0.071 | 0.418 | 0.152 | 0.051 |
| IndexNet [3] | +44.20 | 0.866 | 0.976 | **0.995** | 0.117 | 0.071 | 0.416 | 0.151 | 0.050 |
| A2U [5] | +0.08 | 0.860 | 0.973 | 0.993 | 0.121 | 0.077 | 0.429 | 0.156 | 0.052 |
| SAPA-B | +0.16 | 0.871 | 0.977 | 0.994 | **0.116** | 0.070 | 0.410 | 0.151 | 0.050 |
| SAPA-G | +0.25 | **0.872** | **0.978** | **0.995** | **0.116** | **0.069** | **0.408** | **0.149** | **0.049** |

refer to [11] for the definitions of the metrics. We replace all upsampling stages but the last one for SAPA, due to no available high-res feature map for the last stage.

For image matting, we train the model on the Adobe Image Matting dataset [13] and report four metrics on the Composition-1k test set, including Sum of Absolute Difference (SAD), Mean Squared Error (MSE), Gradient (Grad), and Connectivity (Conn) [29]. A2U matting [5] is adopted as the baseline. We use the code provided by the authors and follow the same training settings as in the original paper.

## 5.2 Main Results

We compare SAPA and its variants against different upsampling operators on the three strong segmentation baselines. Results are shown in Table 2, from which we can see that SAPA consistently outperforms other upsampling operators. Note that SAPA can work well even without parameters and achieves the best performance with only few additional #Params and #FLOPs.

Table 5: Image matting results on Adobe Composition-1k data set. Best performance is in **boldface**.

| A2U Matting [5] (R34) | Params | SAD↓ | MSE↓ | Grad↓ | Conn↓ |
|---|---|---|---|---|---|
| Bilinear | 8.05 | 34.22 | 0.0090 | 17.50 | 31.55 |
| CARAFE [1] | +0.26 | 32.50 | 0.0086 | 16.36 | 30.35 |
| IndexNet [3] | +12.26 | 33.36 | 0.0086 | 16.17 | 30.62 |
| A2U [5] | +0.02 | 32.05 | 0.0081 | 15.49 | 29.21 |
| SAPA-I | +0 | 34.25 | 0.0091 | 18.93 | 32.09 |
| SAPA-B | +0.04 | 31.19 | 0.0079 | **15.48** | 28.30 |
| SAPA-G | +0.04 | **30.98** | **0.0077** | 15.59 | **27.96** |

Table 6: Ablation studies. **I**: inner-product similarity; **B**: bilinear similarity; **P**: plain addition; **G**: gated bilinear similarity.

| SegFormer B1 | Sim func | | | | Embedding dim | | | |
|---|---|---|---|---|---|---|---|---|
| Setting | **I** | **B** | **P** | **G** | 16 | 32 | 64 | 128 |
| mIoU | 43.1 | 43.2 | 43.2 | 44.4 | 43.0 | 43.2 | 43.2 | 43.4 |

| SegFormer B1 | Norm func | | | | | Kernel size | | |
|---|---|---|---|---|---|---|---|---|
| Setting | None | $e^x$ | relu | sigmoid | softplus | 3 | 5 | 7 |
| mIoU | 41.5 | 43.2 | 40.8 | 42.8 | 42.7 | 43.1 | 43.2 | 42.5 |

Results on other three dense prediction tasks are shown in Tables 3, 4, and 5. SAPA outperforms other upsampling operators on depth estimation and image matting, but falls behind CARAFE on object detection. One plausible explanation is that the demand of details in object detection is low (Section 6 presents a further in-depth analysis). In detail-sensitive tasks like image matting, SAPA significantly outperforms CARAFE. Qualitative comparisons are shown in Fig 4.

### 5.3 Ablation Study

Here we conduct ablation studies to compare the choices of similarity function and normalization function, the effect of different kernel sizes, and the number of embedding dimension. For the default setting, we use the bilinear similarity function, apply the normalization function $h(x) = e^x$, set the kernel size $K = 5$ and the embedding dimension $d = 32$. Quantitative results are shown in Table 6.

**Similarity Function**    We investigate three types of similarity function aforementioned and an additional 'plain addition' baseline. It ablates the gating mechanism from the gated bilinear similarity. Among them, gated bilinear similarity generates the best performance, which highlights the complementarity of semantic regions and boundary details in kernel generation.

**Normalization Function**    We also investigate different normalization functions. 'None' indicates no normalization is used. Among validated functions, $h(x) = e^x$ works the best, which means normalization matters. The other three have to play with the *epsilon* trick to prevent zero division.

**Kernel Size**    The kernel size controls the region that each point in the upsampled feature can attend to. Results show that, compared with a large kernel, a small local window is sufficient to distinguish the semantic clusters.

**Embedding Dimension**    We further study the influence of embedding dimension in the range of 16, 32, 64, and 128. Interestingly, results suggest that SAPA is not sensitive to the embedding dimension. This also verifies that SAPA extracts existing knowledge rather than learn unknown knowledge.

## 6 Discussion

**Understanding SAPA from a Backward Perspective**    We have explained SAPA in the forward pass, here we further discuss how it may work during training. In fact, originally the model does not know to which the semantic cluster an encoder point should belong. The working mechanism

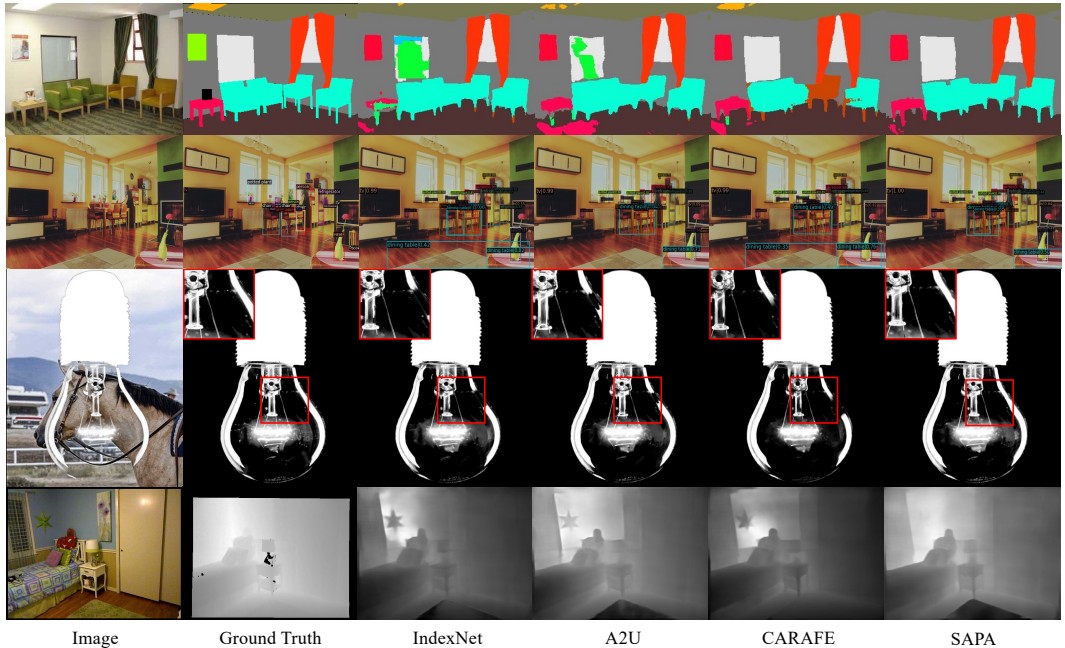

| Image | Ground Truth | IndexNet | A2U | CARAFE | SAPA |

Figure 4: **Qualitative results of different upsampling operators on different tasks.** From top to bottom, semantic segmentation, object detection, image matting, and depth estimation.

of SAPA assigns each encoder feature point a possibility to choose a cluster. During training, the ground truths produce gradients, thus changes the assignment possibility. In SAPA, for every encoder point, the semantic clusters in the corresponding local window in decoder features serve as *implicit labels* and cooperate with the ground truths. The correct cluster is the positive label, while the wrong one is negative. In the preliminary stage of training, if an encoder feature point is more similar to a wrong cluster, it will be punished by gradients and engender large losses, and vice versa. Therefore, the encoder feature points can gradually learn its affiliation. We find this process is fast by visualizing the epoch-to-epoch feature maps.

**Limitations compared with CARAFE** CARAFE is a purely semantic-driven upsampling operator able to mend semantic information with a single-input flow. Such a mechanism in CARAFE brings advantages in smoothing semantic regions. *E.g.*, we observe it mends holes in a continuous region. However, due to its single-input flow, it cannot compensate the details lost in downsampling. Our SAPA, by contrast, mainly aims to compensate details such as textures and boundaries. SAPA characters in two aspects: semantic preservation and detail delineation. However, as Eq. (4) suggests, we do not add any semantic mending mechanism in SAPA. This explains why SAPA is worse than CARAFE on object detection, because detection has less demand for details but more for regional integrity. In short, CARAFE highlights *semantic mending*, while SAPA highlights *semantic preserving* and *detail delineation*.

## 7   Conclusion

In this paper, we introduce similarity-aware point affiliation, *i.e.*, SAPA. It not only indicates a lightweight but effective upsampling operator suitable for tasks like semantic segmentation, but also expresses a high-level concept that characterizes feature upsampling. SAPA can serve as an universal substitution for conventional upsampling operators. Experiments show the effectiveness of SAPA and also indicate its limitation: it is more suitable for tasks that favor detail delineation.

**Acknowledgement**   This work is supported by the Natural Science Foundation of China under Grant No. 62106080.

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
