# SAPA: Similarity-Aware Point Affiliation for Feature Upsampling
# Supplementary Materials

**Hao Lu    Wenze Liu    Zixuan Ye    Hongtao Fu    Yuliang Liu    Zhiguo Cao**[*]

School of Artificial Intelligence and Automation
Huazhong University of Science and Technology
Wuhan 430074, China
{hlu,zgcao}@hust.edu.cn

We provide the following contents in this supplementary:

- Output visualizations of different upsampling operators on some reported tasks for qualitative comparison;

- Additional visualizations of the intermediate process of SAPA;

- Implementation details of reported experiments;

- CUDA implementation for SAPA.

## S1    Output visualizations of different upsampling operators on reported tasks

We first visualize the qualitative results of different upsampling operators. Note that we do not provide visualizations for object detection here, because we do not observe significant differences between difference upsampling operators. For semantic segmentation, we visualize the segmentation masks of SegFormer. It involves 6 upsampling stages in all and thus can reveal the differences of upsampling operators more clearly.

Semantic segmentation visualizations are shown in Fig. S1, image matting ones are shown in Fig. S2, and visualizations of monocular depth estimation are shown in Fig. S3.

## S2    Additional visualizations of the intermediate process of SAPA

To better understand how SAPA works, we supplement additional visualizations on the encoder features, decoder features, upsampling kernels, and upsampled features of SAPA. As shown in Fig. S4, for the upsampling kernels, we choose every top-left weight of the upsampling kernel for visualization, therefore the kernel map is of the same size with the upsampled feature. From the kernel map we can see that the top-left kernel weights mainly response to top and left edges, which explains the affiliation of the top-left point among the upsampled four. Additionally, we use bilinear interpolation to upsample the same decoder feature for a visual comparison, and the results are shown at the rightmost column. We can see that bilinear interpolation generates more blurry feature maps than SAPA. This vague is more harmful to features of low resolution. Therefore, by replacing the upsampling operator in every upsampling stage, SAPA can manifest stronger semantic preservation and boundary delineation than other competitors.

---

[*]Corresponding author.

36th Conference on Neural Information Processing Systems (NeurIPS 2022).

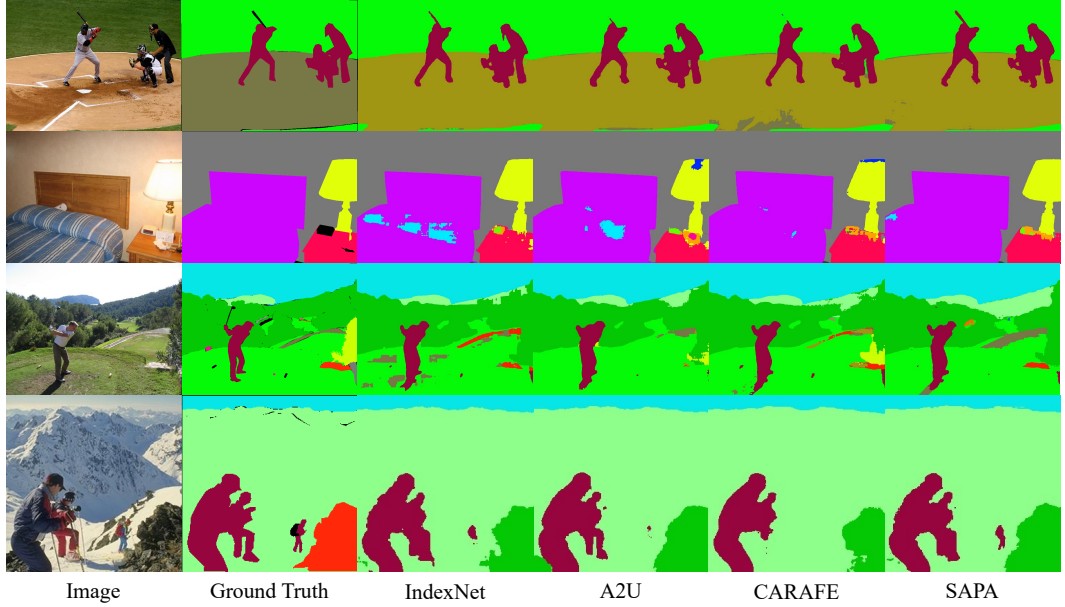

| Image | Ground Truth | IndexNet | A2U | CARAFE | SAPA |

Figure S1: Visualizations of different upsampling operators on semantic segmentation with SegNet-B1 as the baseline.

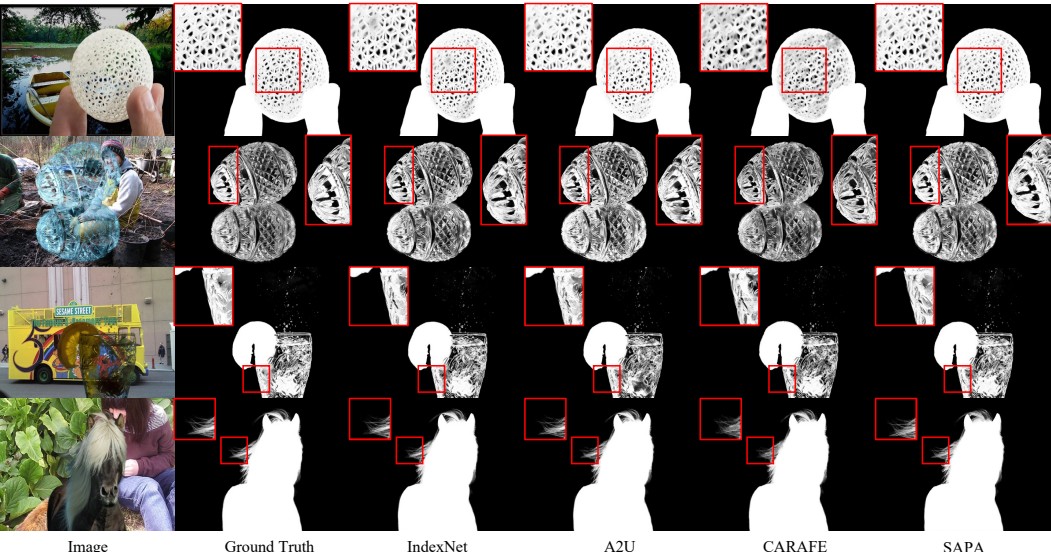

| Image | Ground Truth | IndexNet | A2U | CARAFE | SAPA |

Figure S2: Visualizations of different upsampling operators on image matting with A2U as the baseline.

## S3    Implementation details

**Intermediate visualization with SegNet on SUN RGBD**    SegNet is a relatively simple baseline such that we can focus on the effect of upsampling with less disturbance. Therefore we select SegNet as the baseline model and replace all 5 upsampling operators to investigate the working mechanism of SAPA. We use the SUNRGBD dataset and adopt random scaling, random cropping and random flipping in data augmentation. The backbone VGG-16 is pretrained on ImageNet. The batch size is set to 8. We use the cross entropy loss and the SGD optimizer with a $0.9$ momentum. The model is trained for 200 epochs with a constant learning rate of $0.01$.

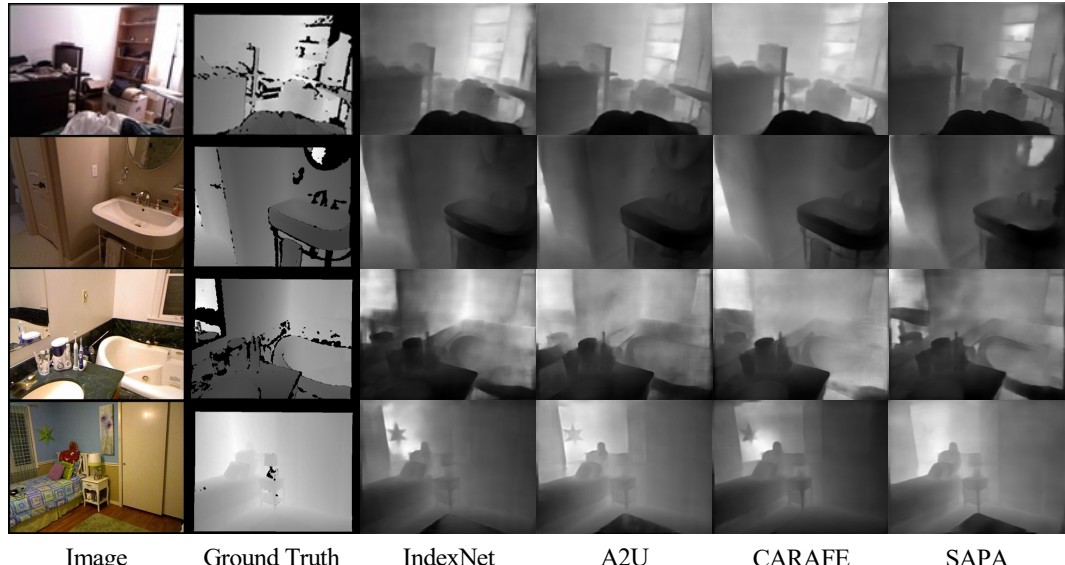

| Image | Ground Truth | IndexNet | A2U | CARAFE | SAPA |

Figure S3: Visualizations of different upsampling operators on depth estimation with BTS as the baseline.

**Semantic Segmentation on ADE20K**   We use the codes released by the authors. We keep all other settings unchanged while only modify the upsampling stages. For SegFormer[2], we use the B1 baseline. In SegFormer, the feature maps of 3 different scales are finally concatenated, and we apply $\times 2$ upsampling for $3 + 2 + 1 = 6$ times. For MaskFormer[3] and Mask2Former[4] with the FPN architecture, we use Swin-Base model pretrained on ImageNet-22K as the backbone, and they have 3 and 1 upsampling stages, respectively.

**Object Detection on MS COCO**   We use the Faster R-CNN model implemented by `mmdetection`[5]. We keep all settings unchanged and only modify the upsampling stages in FPN.

**Monocular Depth Estimation on NYU Depth V2**   We use the code provided by BTS[6] and set the batch size as 4. Considering that the last upsampling stage does not have a guided high-resolutin encoder feature map, we only modify the other stages when comparing different upsampling operators. In this model the encoder and decoder feature dimensions are different, so SAPA-I is not supported.

**Image Matting on Adobe Composition-1K**   We use the code provided by A2U matting[7]. Max-pooling with kernel size 2 and stride 2 is used at all downsampling stages. And then we modify only the upsampling stages.

## S4   CUDA Implementation for SAPA

Since there is no standard library in PyTorch that supports our operations. If PyTorch is used, the `unfold` function must be involved, which can bring much memory cost. Therefore, we provide a CUDA-based implementation, which can be found at https://github.com/poppinace/sapa. We also provide the equivalent PyTorch code for reference, but all our experiments are conducted based on the CUDA implementation.

---

[2]https://github.com/NVlabs/SegFormer

[3]https://github.com/facebookresearch/MaskFormer

[4]https://github.com/facebookresearch/Mask2Former

[5]https://github.com/open-mmlab/mmdetection

[6]https://github.com/cleinc/bts

[7]https://github.com/dongdong93/a2u_matting

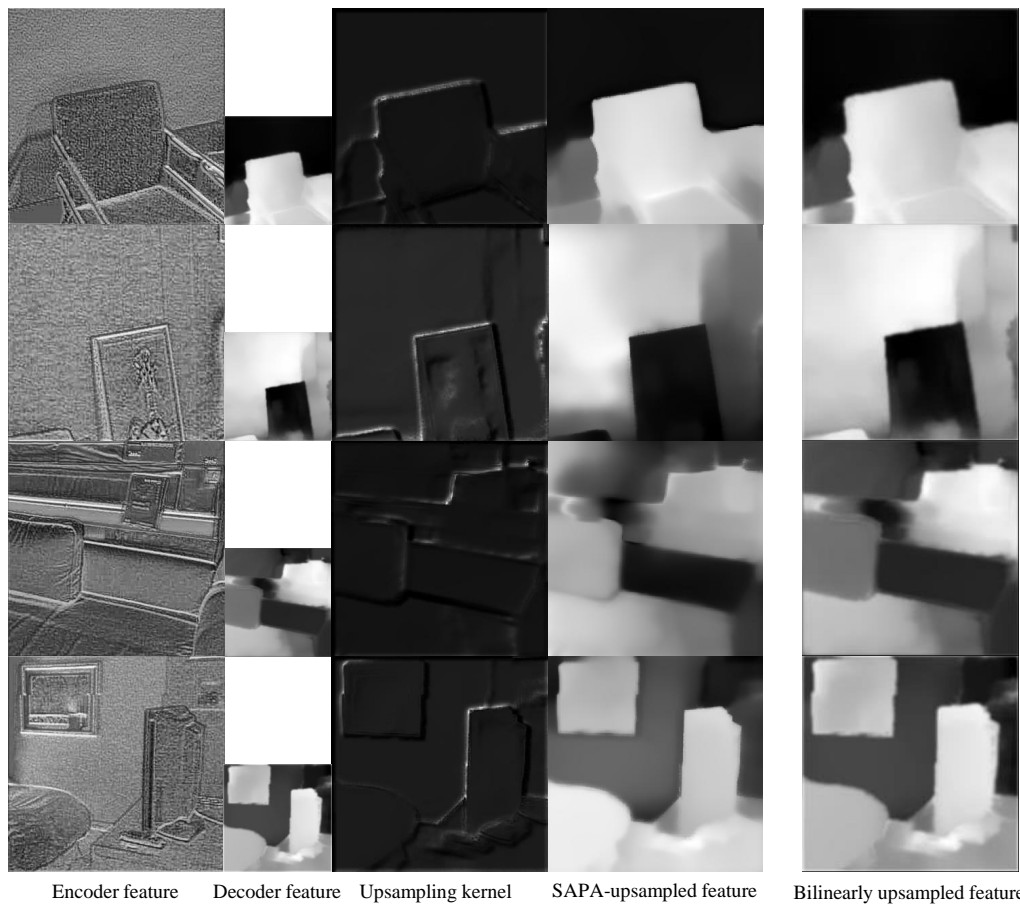

Encoder feature     Decoder feature     Upsampling kernel     SAPA-upsampled feature     Bilinearly upsampled feature

Figure S4: Visualizations of intermediate processes of SAPA.