# OpenReview forum: "SAPA: Similarity-Aware Point Affiliation for Feature Upsampling"
_NeurIPS.cc/2022/Conference — NeurIPS 2022 Accept_

### Official Review · Reviewer_jFju · 2022-06-30

**Rating:** 6
**Confidence:** 4
**Soundness:** 4 excellent
**Presentation:** 3 good
**Contribution:** 3 good

**Summary:**

This paper introduces a new kernel upsampling module which encourages not only semantic smoothness but also boundary sharpness in order to enhance the performance of semantic segmentation/matting tasks. The key idea is to design a similarity-aware kernel which compare the similarity between encoded features with local spatial awareness about the decoded features. The paper has also proposed a lightweight upsampling operator,  Similarity-Aware Point Affiliation (SAPA), which demonstrated high performance on various dense prediction tasks. The proposed method has compared with CARAFE, IndexNet and A2U in term of both quantitative results and computational complexity.

**Questions:**

Although table 1 has summarize the theoretical complexity of various, I would like to see the actual runtime of the proposed method compared with other methods.

Besides the quantitative results in table 2 and table 3, I would like to see some qualitative results which show the boundaries of the segmented regions, especially for the matting tasks.

The ablation study about the kernel size seems to be too simple, I hope there is a deeper study about its effectiveness.


**Limitations:**

The paper has discussed it properly.

**Strengths And Weaknesses:**

Overall, this is a solid paper and I think the proposed method is very useful in dense prediction tasks that require very accurate boundary. The experimental comparisons are also sufficient to validate the performance of the proposed method, and I think its simplicity would make this method useful in real world applications.

---

> ### Author Response · Authors · 2022-07-31
> **Response to Reviewer jFju**
>
> We thank the reviewer for positive comments and consider our approach useful. We answer the questions as follows.
>
> **Actual runtime comparison.**
>
> We test the runtime on a single NVIDIA GeForce RTX 1080Ti GPU with Intel Xeon CPU E5-1620 v4 @ 3.50GHz CPU.
> By upsampling a random feature map of size 256\*120\*120 to 256\*240\*240, CARAFE, IndexNet, A2U, and SAPA-B
> takes 2.75 ms, 7.39 ms, 8.53 ms and 6.24 ms, respectively (averaged over 1000 independent runs).
> Considering that SAPA processes five times more data than CARAFE (due to the high-res encoder features),
> we think the time consumption of SAPA is acceptable.
>
> **More qualitative results.**
>
> Due to the limited capacity of main text, we have moved the qualitative results to the supplementary material.
> Please refer to Figure S1-S3 in the supplementary material for the visualizations on the reported task.
>
> **Ablation study on kernel size.**
>
> Indeed it seems that the ablation study on kernel size is too simple.
> Besides the quantitative experiment results, we can explain more why we choose $K=5$ here.
> Considering the situation where the boundary in the low-res feature is not that sharp,
> there are likely gradually changing values on the boundary, where one cannot see distinct semantic clusters in a small window such as $3\times 3$.
> On the other hand, because our used normalization function $h(x)=e^x>0$, i.e., every kernel weight is larger than zero,
> if too large kernel size is chosen, the smoothing effect of the kernel increases, which has also a negative influence on sharp boundary.
> Therefore, by considering all these factors and the ablation study results, we choose the kernel size as $5$.
> We will also investigate more on our operator, perhaps in a journal extension due to the limited capacity of the conference paper.

---

### Official Review · Reviewer_4pwd · 2022-07-11

**Rating:** 5
**Confidence:** 4
**Soundness:** 2 fair
**Presentation:** 2 fair
**Contribution:** 2 fair

**Summary:**

This work proposes a new approach for upsampling decoder features with the guidance of encoder feature, leading to semantic preserving and detail delineation. The key idea is to apply similarity-aware upsampling based on encoder and decoder features. To be more specific, the decoder feature is upsampled via the weighted sum in the sub-region, where the weight is calculated using the similarity between encoder and decoder features, as shown in Figure 3.

**Questions:**

- In Figure 3, the decoder feature is added into the encoder feature. This was not explained in the paper.

- The similarity kernel w_{i1, j1, m, n}, which is used for obtaining the upsampled decoder feature at (i1, j1), is calculated with the single encoder feature at (i1, j1) and the set of decoder features (m,n) for m=-r,...,r and n=-r,...,r. This means that upsampling relies on the single encoder feature at (i1, j1), and thus it seems that the proposed method is still too sensitive to the noise of the encoder feature.


**Limitations:**

N.A.

**Strengths And Weaknesses:**

+) The dense prediction tasks (semantic segmenation, depth etimation, and image matting) often require an accurate and detail-preserving upsampling of decoder features, and the proposed simple upsampling scheme can be very effective with no significant computational overhead.

+) Experiments validated the performance gain on the above-mentioned tasks.

-) The overall idea is very simple, and it can be explained using (2) and (3) together with Figure 3. Nevertheless, the manuscript needs to be revised in a more compact form. In the abstract and introduction, authors stated that the point affiliation (semantic cluster) should be incorporated in the upsampling process of decoder features. 'Cluster' seems to be rather misleading. The proposed idea is just a similarity based summation, where the similarity is measured between encoder and decoder features. There is no explicit clustering and affiliation estimation process in the proposed framework.

-) The equation (4) is rather meaningless. The proposed kernel is rather similar to joint bilateral filter, and it is well-known that in the smooth region, the joint bilateral kernel becomes the Gaussian function. Though the proposed kernel becomes a uniform function, such a derivation seems unnecessary in the main paper (maybe it can be moved to supplementary material).

**********************************************************************************
* After rebuttal

Authors addressed some questions in the rebuttal. I appreciate it.

This work introduces a simple yet effective approach for dense prediction tasks. This kind of upsampling (using two inputs) has been adopted in various vision tasks, but using it in the upsamling process for decoder features sounds interesting.
Nevertheless, considering the simplicity of the overall method, I would like to keep the initial rating (Borderline accept).
**********************************************************************************

---

> ### Author Response · Authors · 2022-07-31
> **Response to Reviewer 4pwd**
>
> The authors thank the reviewer for constructive comments, particularly on presentation. We address these concerns as follows.
>
> **The notion of 'cluster' seems misleading. There is no explicit clustering and affiliation estimation process in the proposed framework.
> The manuscript needs to be revised in a more compact form.**
>
> Our approach is not related to clustering approaches. We use the term 'semantic cluster' to indicate a region where points have similar semantic meaning.
> Since this term has been clearly defined in the footnote of page 1, we think this may not mislead readers.
> In addition, we do have an affiliation assignment process, but in an implicit way with the similarity scores in the kernel.
> By encoding the mutual similarity between encoder and decoder feature points in the kernel,
> the upsampled point could be assigned to a semantic cluster that is most similar to.
>
> It is true that our idea can be explained with Eq. (2), Eq. (3), and Figure 3, but we think other parts can help one to understand our idea more easily.
> Following the suggestion of the reviewer, we have simplified the symbol system and have rewritten some parts of text to improve the clarity and conciseness,
> e.g., we have squeezed Eq. (4) into one single line. Please take a look at our submitted revision.
>
> **The equation (4) is meaningless.**
>
> In contrast to the reviewer, we think that Eq. (4) expresses a key characteristic of SAPA for noise suppression on the encoder feature.
> Our work originates from an observation that upsamplers using encoder features like IndexNet and A2U work poorly in semantic segmentation.
> From Fig. 2, their upsampling kernels introduce unexpected texture or color from encoder features into upsampled features, which affects semantic coherence.
> Yet, involving encoder features does enhance the boundary quality,
> so wezh explore how to use encoder features to compensate details while not introducing noise, especially on interior regions.
> Eq.(4) and the comparison in Figure 2 exactly explain and emphasize how we effectively block the noise from the encoder features.
> Hence, we expect to leave Eq. (4) as it is.
>
> **The kernel is similar to Joint Bilateral Filter (JBF).**
>
> In JBF, $J_p=\frac{1}{k_p}\sum\limits_{q\in\Omega}I_qf(||p-q||)g(||\widetilde{I}_p-\widetilde{I}_q||)$, where $f$ is the spatial filter kernel,
> such as a Gaussian centered over $p$, and $g$ is the range filter kernel conditioned on a second guidance image $\widetilde{I}$,
> centered at its image value at $p$. $\Omega$ is the spatial support of the kernel $f$, and $k_p$ is a normalizing factor.
>
> From the formulation above and Eq.(3) in our paper we see that SAPA and JBF actually are not similar but differ in:
>
> - **The way to generate kernels.** JBF uses the product of two kernels&mdash;a spatial filter conditioned on the distance prior of $I$
> and a range filter conditioned on $\widetilde{I}$&mdash;to generate the kernel, while SAPA only generate one kernel with mutual similarity.
>
> - **The source of similarity comparison.** In the range filter, JBF calculates the similarity only between the points in the higher-res feature
> $\widetilde{I}$; however, SAPA computes the similarity of each high-res point in the encoder and its corresponding low-res decoder points within a window.
>
> In JBF, when the higher-res feature point is in smooth regions, i.e., $\widetilde{I}_p=\widetilde{I}_q$, then the kernel becomes Gaussian.
> But in our paper, we discuss: when the low-res feature points are in a smooth region, i.e., $I_p=I_q$, how to retain its smoothness.
> Considering the texture noise, which results in $\widetilde{I}_p\neq\widetilde{I}_q$, if JBF is used, the kernel will not be a Gaussian,
> and will even be sensitive to the texture noise. However, in this case, SAPA (Eq. (4)) enables the kernel to keep an unchanged constant,
> regardless of the value of the encoder point.
>
> **The explanation of Fig. 3.**
>
> We are sorry for causing misleading here. SAPA has three variants, named SAPA-I, SAPA-B, and SAPA-G.
> We attempt to merge the three in a single figure, but it seems rather confusing.
> In Figure 3, the addition symbol and the switch indicate our gated bilinear similarity version named SAPA-G.
> We have found a way to improve the clarity of Fig. 3 and have updated it in the revision.
>
> **The sensitivity to the noise in the encoder feature.**
>
> As mentioned above, noise suppression does not rely only on a single encoder point,
> but on the similarity between each encoder point and its corresponding local decoder points.
> Per Eq. (2) and Eq. (4), SAPA will upsample a smooth region to a smooth region, regardless of the value of the encoder point.
> Note that, we refer the 'noise' as the unexpected details in encoder features compared with the label mask.
> If segmenting a person from background, then the clothes on body would be noise.
> We do not refer to the signal noise that may destroy the image content. We assume we still process natural images.

---

### Official Review · Reviewer_kfqX · 2022-07-11

**Rating:** 5
**Confidence:** 3
**Soundness:** 3 good
**Presentation:** 3 good
**Contribution:** 3 good

**Summary:**

The paper presents an interesting feature unsampling module, which can be flexible applied to the tasks with upsampling like segmentaiton, detection and depth estimation. The main idea is to generate the kernels based on the feature clustering similarity. It provides extensive experiments on different dense prediction tasks and consistent performance gain has been obtained. The paper is well motivated and the presentation is clear.

**Questions:**

1. The segmentation experiments are based on three transformer-based models. How about the segmentation performance for the CNN-based approaches?
2. The proposed approach relies on the similarity score. It is similar to a new interpolation between the features. Thus, how about is result of the high-frequency data if the ground-truth mask within a  local neighborhood changes abruptly?

**Limitations:**

The paper claims the potential limitation on the object detections which requires on the semantic mending.

**Strengths And Weaknesses:**

strength:
1. The idea to design a feature upsampling framework based on the clustering similarity is interesting and novel.
2. The proposed module obtains state-of-art performance on dense prediction tasks like segmentation, detection, and depth estimation, without large computational overhead.
3. The proposed module is well motivated and the presentation of the paper is clear.

weakness:
1. As shown in Table 3, the performance of on object detection is a little bit lower than the baseline of CARAFE [1]. Although the authors provide an explaination in Section 6, usually the better segmentation results should lead to better localization on the bounding-box level.
2. Compared with the baselines, the implementation of the proposed approach is a little bit complicated with more steps.

---

> ### Author Response · Authors · 2022-07-31
> **Response to Reviewer kfqX**
>
> We thank the reviewer for considering our work novel and interesting. We address the questions and concerns as follows.
>
> **Lower performance than CARAFE on object detection.**
>
> The AP metric of object detection can be influenced by both classification and localization;
> CARAFE mainly improves misclassification due to the ability of semantic mending, for instance, by reducing false positives;
> however, when the classification is not solved well, the advantage of better localization introduced by SAPA can be marginal.
> Such a difference comes from the different emphases of the two upsampling operators.
>
> **The proposed approach has a little bit more steps compared with baselines.**
>
> In Table 1, we ignore the steps of other upsamplers in generating kernels, and depict ours in detail,
> so it seems that ours is more complicated. Actually, our implementation steps are similar to the compared upsamplers;
> if the gating mechanism is not considered, SAPA is even more concise.
> We add the gating mechanism because this additional step brings considerable increase of performance.
> Moreover, SAPA can achieve good performance even without the gating mechanism.
>
> **Segmentation performance on the CNN-based approaches.**
>
> We select the three transformer-based baselines because they are the recent mainstream in semantic segmentation.
> SAPA is applicable to CNN models as well. To prove this, here we supplement an experiment with UperNet on ADE20K dataset with ResNet-50 (R50) as the backbone.
> We train UperNet-R50, with upsamplers in FPN replaced by SAPA-B, for 80 K iterations, and reach the mIoU of 41.47,
> which outperforms the original bilinear baseline by 0.77.
>
> **Dealing with high-frequency data within a local neighborhood.**
>
> The high-frequency neighborhood follows the same principle as the low-frequency neighborhood; the former may result in additional semantic clusters,
> but the assignment of point affiliation still obeys the same rules given in the paper. Due to the complexity in expressing this graphically,
> we only use the case of two clusters as an example in the paper. One empirical evidence we can offer here is the evaluation on the matting task,
> where ground-truth alpha mattes contain many high-frequency local regions; SAPA still invites consistent performance improvements in all metrics.

---

### Official Review · Reviewer_W9Gs · 2022-07-12

**Rating:** 5
**Confidence:** 3
**Soundness:** 3 good
**Presentation:** 2 fair
**Contribution:** 3 good

**Summary:**

This paper introduces a point affiliation for feature upsampling which is one of the most essential parts, especially dense prediction networks. The proposed method generates similarity-aware kernels by comparing the similarity between each encoder feature point and the spatially associated local region of decoder features. It also introduce a lightweight upsampling operator, termed Similarity-Aware Point Affiliation (SAPA) and its variant. Experiments show the superiority of the proposed upsampling module on various depth prediction tasks.

**Questions:**

- Learning feature upsampling is interesting and makes sense. But this framework requires additional learnable parameters, which may make the networks more suffer from the overfitting problem, less generalizable. It would be great if there are experiments about this overfitting issue and generalization issues.
- In Kernel generation part, the similarity between encoder feature and decoder feature is computed. But, such encoder feature and decoder feature may have different data distributions, so directly computing the similarity between them may be sub-optimal. To overcome this, very simple normalization prior to computing similarity may be used. It would be great if the relevant comments or experiments are additional conducted.
- The proposed module, SAPA, consists of many sub-modules, e.g., Y embedding, X embedding, Gated addition, Kernel generation, etc. So through experiments for ablation study is required. The current ablation study in Table 4 did not cover all the variants of ablation study.

Minor comments:
- Using subscript of subscript, e.g., in Line 184, may make the reader follow the paper. It would be better if simpler notations are used.
- In Similarity Function in Line 203-205, why "low-rank" versions are used? Please clarify this.

**Ethics Review Area:**

["I don’t know"]

**Limitations:**

The paper discussed the limitation of the proposed method.

**Strengths And Weaknesses:**

+ Introducing the notion of point affiliation into feature upsampling is interesting and makes sense. Many other followers will benefit from such notions.
+ In depth comparison to other previous upsampling methods such as CARAFE or IndexNet, e.g., in Fig. 2, is interesting.
+ In most cases, the state-of-the-art performance is attained when it is incorporated with various dense prediction networks.

---

> ### Author Response · Authors · 2022-07-31
> **Response to Reviewer W9Gs**
>
> We appreciate the reviewer for highlighting the significance of point affiliation. We address the concerns as follows.
>
> **Learnable upsampling parameters may lead to overfitting.**
>
> We address this concern from three aspects:
>
> 1) Our framework only introduces a few amount of additional parameters, which occupies $0.03\%\sim1.4\%$ of the overall number of parameters in the baseline models.
> Even if a model overfits data due to excessive number of parameters, perhaps the baseline model should be checked first.
> It is unlikely that such a few additional upsampling parameters would dominate overfitting.
>
> 2) SAPA-I has no additional parameter. The results in Table 2 show that SAPA-I still achieves good performance.
>
> 3) Overfitting may loosely related to the number of upsampling parameters but a specific upsampler used.
> We use a toy-level experiment to exposit this claim.
> Since overfitting happens more likely on small data sets, we select a small two-class segmentation dataset, the Weizmann Horse,
> and use SegNet as the baseline. We replace the default max unpooling with NN interpolation, bilinear interpolation, and SAPA-B, respectively.
> We train the model for 50 epochs and use a constant learning rate of 0.01.
> By plotting the learning curves, the training losses of the three upsamplers all decrease smoothly to $0.26\sim0.36$,
> but their val losses vary significantly.
> At the 10-th epoch the val loss of SegNet-NN begins to increase, from the minimum of 0.126 to 0.166 at the 50-th epoch,
> and that of SegNet-bilinear increasees at the 14-th epoch, from 0.119 to 0.138 at the 50th epoch.
> Instead, the training loss of SegNet-SAPA decreases faster and the val loss reaches the minimum of 0.057 within 10 epochs,
> and fluctuates from 0.057 to 0.060 during the rest of epochs. The mIoU metrics are 89.5 for NN, 90.1 for bilinear, and 94.9 for SAPA.
> The experiment shows that the additional learnable parameters in SAPA encourage not only fast training but also better generalization.
> Hence overfitting may have weak correlation with learnable upsampling parameters.
> Perhaps as Bengio's ICLR 2017 paper says, "understanding deep learning requires rethinking generalization".
>
> **Normalization should be used before similarity computation due to different data distribution between encoder and decoder features.**
>
> We do have a LayerNorm used before similarity computation (see our code implementation in the supplementary).
> We thank the reviewer for reminding us of this detail that has been overlooked in the submission.
> Indeed, without normalization, the network even cannot be trained.
> We have supplemented this detail in the revision (L211-L214, "In practice, encoder and decoder features may have different data distributions,
> which is not suitable to compare similarity directly. Therefore we apply $\tt LayerNorm$ for both encoder and decoder features before similarity computation.
> Indeed we observe that the loss does not decrease without normalization").
>
> **The current ablation study in Table 4 did not cover all the variants.**
>
> We think the ablation study in Table 4 has included most circumstances mentioned by the reviewer.
> SAPA has two modules: kernel generation and feature assembly. Feature assembly is a standard procedure and does not require ablation study.
> The kernel generation has 3 steps: XY embedding, similarity computation, and kernel normalization.
>
> - **XY embedding.** The effect of embedding or not can be observed by comparing SAPA-I and SAPA-B, and the results are shown in Tables 2, 3, and 4.
> Additionally, we also explored the influence of embedding dimension in Table 4.
>
> - **Similarity computation.** We have validated inner product, bilinear, and gated bilinear similarity in main experiments (Tables 2 and 3).
> Then, because gated bilinear similarity follows a gated addition manner, in Table 4 we further included a plain addition baseline (P).
> By comparing P and G in Table 4, it also justified the effectiveness of gated addition.
>
> - **kernel normalization.** We have explored four normalization functions for computing the similarity score in Table 4.
> Additionally, we also explored the influence of the upsampling kernel size.
>
> Perhaps all the results are summarized in a single line in Table 4, which makes it difficult to interpret the results.
> We have reorganized Table 4 and double checked what is missing in the ablation studies.
> We further supplement the result of the upsampling kernel without normalization (41.45 mIoU).
> See the Table 4 of the revision (L270-L272).
>
> **The use of the "low-rank" version.**
>
> The motivation behind low-rank version is to reduce the number of parameters and computational complexity.
> We have clarified this in the revision (L200-L202).
>
> **Using subscript of subscript should be avoided.**
>
> We have simplified the symbol system to improve the readability. Please have a look at the revision.

---

### Meta-Review · Area_Chair_DZLo · 2022-08-26

**Recommendation:** Accept
**Confidence:** Less certain

**Metareview:**

The paper focuses on the task of feature upsampling, specifically in decoder layers for dense prediction problems. The proposed point affiliation module can be used in upsampling kernels to produce semantically smooth and boundary preserving upsampled sets. The paper received four detailed reviewers from experts. There was a healthy discussion between authors and reviewers during the discussion period and the extra analyses, explanation, and experiments from the authors helped resolve most of the concerns raised by the reviewers. With these extra items presented in the discussion period, the paper has reached the level of impact and contribution expected by NeurIPS papers. The authors are recommended to include them in the final version of the paper.

**Award:**

No

---

### Decision · Program_Chairs · 2022-09-14

Accept